# Corporate Social Responsibility and SMEs in Vietnam: A Study in the Textile and Garment Industry

**Loan Thi-Hong Van** [1],* and **Phuong Anh Nguyen** [2]

1   School of Advanced Study, Ho Chi Minh City Open University, Ho Chi Minh City 7000, Vietnam
2   Graduate School, Ho Chi Minh City Open University, Ho Chi Minh City 7000, Vietnam;
    anhphuong262@gmail.com
*   Correspondence: loan.vth@ou.edu.vn

**Abstract:** This study explored the influence of factors on the implementation of corporate social responsibility (CSR) in companies. The study used a quantitative approach in which a survey was conducted. The final 250 among various respondents in the textile and garment industry were used. The final respondents were top-, middle-, and low-level managers in 250 small and medium enterprises (SMEs) in Vietnam. The results indicate that competitive context, social influences, the understanding of managers about CSR, and the internal environment of companies are the four drivers of CSR. In the four drivers, competitive context has the strongest impact on adopting CSR. The finding implies that stakeholders' pressure influences SMEs in this industry because of the high expectations from international stakeholders.

**Keywords:** corporate social responsibility; textile and garment industry; Vietnam

## 1. Introduction

Corporate social responsibility (CSR) has had a long history of development in the world (Carroll 2009). However, it is practiced differently in countries because of the different contexts (Nguyen et al. 2017; Omran and Ramdhony 2015; Gupta 2009). Most CSR studies take place in developed countries; thus, there is a need to examine CSR in developing countries (Eweje 2006). Research about CSR in Vietnam, a developing country, is sparse (Nguyen et al. 2017, Nguyen and Truong 2016). Current CSR research additionally focuses on large organizations rather than small and medium enterprises (SMEs) (Jenkins 2006; Morsing and Perrini 2009). It is hard to use the theories and practices of large enterprises on SMEs due to SMEs' characteristics (Davies and Crane 2010). Thus, there need to be more studies with findings which are useful and applicable to SMEs.

Research on CSR and SMEs has only recently emerged (Murrillo and Lozano 2006; Morsing and Perrini 2009; Vo 2012). Literature needs more studies on these topics. One of the research gaps is the engagement of CSR in SMEs (Murrillo and Lozano 2006; Vo 2012). Research is needed to explore why SMEs adopt CSR. Some researchers state that not many SMEs conduct CSR due to the lack of resources (see, e.g., Lepoutre and Heene 2006; Kusyk and Lozano 2007; Sweeney 2007; Nguyen and Pham 2016), or the lack of CSR understanding (Tran and Jeppesen 2016). Others say that SMEs implement CSR because of ethical reasons (Longo et al. 2005; Jenkins 2006), business performance and regulation (Williamson et al. 2006), relationship with the community and company image improvement (Longo et al. 2005), and capital and human resources (Nguyen and Pham 2016).

CSR was introduced in Vietnam in recent years (Hamm 2012; Tran and Jeppesen 2016), although its definition by Bowen was published in 1953 (Carroll 1979). Many Vietnamese companies have therefore had difficulties in adopting the concept due to their little CSR knowledge. For example, from January to May 2019, European countries refused 17 shipments of seafood from Vietnam because of

the lack of knowledge relating to food safety and hygiene (Bach 2019). The wastewater emissions by the Formosa Plastics Corporation in the Ha Tinh province in 2016, and the food scandal by the Tan Hiep Phat Beverage Group in 2015, are other examples of the limitations of CSR knowledge. The disapprobation of stakeholders to these companies implies that stakeholders have taken notice of social responsibilities in spite of the fact that CSR is new to them (see Phương 2015; Quang et al. 2016; Nguyen and Truong 2016).

According to a survey by the General Statistics Office of General Statistics Office of Vietnam (2017), a large majority of companies in Vietnam are SMEs (98.1%). In Vietnam, SMEs are companies whose charter capital is below 10 billion Vietnam dong ($430,000) and whose number of employees is lower than 300 (Vietnamese Government 2001). Vietnam's textile and garment industry contributes to 10 percent of the national industrial values and creates 2.7 million jobs—this shows the importance of the industry in the Vietnamese economy. Additionally, a characteristic of the industry is that most companies have collaborated with international partners (Song 2018). These are our reasons for selecting the industry for our research. It may help to understand how to promote CSR in SMEs, which is little known (Murrillo and Lozano 2006). SMEs in other industries may not want to practice CSR, but SMEs in the textile and garment industry in Vietnam may do so due to the high expectations of their international stakeholders.

Although the industry exported products worth more than 36 billion US dollars in 2018—reaching the top three in the world, together with China and India (Song 2018)—problems still occurred relating to CSR. For example, in July 2019, the Big C supermarket, a foreign supermarket in Vietnam, announced that it was stopping its collaboration with 200 textile and garment SMEs because of product quality (Kieu 2019). These may show the lack of CSR in their business strategy. In SMEs, owners/managers have the power in their hands to make decisions in personal ways (Spence 1999). Therefore, this study explores the understanding of the CSR of Vietnamese managers in SMEs, in order to understand what CSR means in their companies, and which drivers force them to practice CSR. The purpose of this research was to study factors that affect the implementation of CSR in SMEs in the textile and garment industry in Vietnam. The study used a survey of 250 managers in 250 SMEs in Vietnam in the textile and garment industry. The results indicated that SMEs adopt CSR because of a competitive environment, social influences, the understanding of managers about CSR, and the internal environment of SMEs.

The next section of this article is a review of the existing literature on CSR and SMEs. Next, we elaborate on the collection of the data for the study. Then, we present our research findings and discuss them. The last part is a conclusion in which we show the limitations of the study and the implications for future research.

## 2. Theoretical Background and Hypotheses

CSR was defined as "the commitment of business to contribute to sustainable economic development, working with employees, their families, the local community, and society to improve their quality of life, in ways that are both good for business and good for development" (World Bank 2006). Campbell (2006) defined that "CSR sets a minimum behavioral standard that aims at doing no harm to stakeholders and if it has happened then rectifies it as soon as it is identified." Business companies have to have ethical behaviors, minimize negative influences, and maximize their benefits to society. They need to practice social responsibilities as per the requirements of stakeholders and society.

Garriga and Mele (2004) classified the theories used in CSR research into four groups: instrumental theories, integrative theories, political theories, and ethical theories. The first two groups focus on business management issues, whereas the last two concern society power and ethical codes. A large number of CSR studies use a management perspective, which concerns what motivates business organizations to become engaged in CSR (Basu and Palazzo 2008). One of the groups of integrative theories' stream is stakeholder management that focuses on stakeholder satisfaction. Research by Mitchell et al. (1997) showed that an organization's CSR activities affect its stakeholders' interest and behavior. This also means that the pressure of stakeholders (internal and external) can

motivate organizations to conduct CSR activities. Internal stakeholders include staff, workers, and managers. Their opinions may influence the way to practice CSR in their companies. In other words, managers' ideas and their understanding of CSR can be a driver for CSR implementation in their organizations. External stakeholders can be associations, government agencies, the community, customers, partners, and suppliers (Akkucuk 2015). Most of Vietnam's textile and garment goods are exported, approximately 90 percent of the whole industry (Song 2018; Luu 2018), so international stakeholders may constitute a motive to put CSR into companies' business strategies due to competition—this is one of my research focuses.

Research about CSR was originally undertaken in developed countries (e.g., Carroll 1979, 1991; Carroll 2009); however, the CSR concept is now used in different parts of the world. Visser (2008) said that compared to developed countries, developing countries have different emphases of CSR domains because of "indigenous cultural traditions of philanthropy, business ethics . . . " (p. 481). There is a call to study CSR in developing countries (such as Vietnam), in order to contribute knowledge to world literature (Eweje 2006; Carroll 2016). That is because CSR is influenced by the context where it is applied (Ortenblad 2016; Nguyen et al. 2017). In other words, social expectations about social responsibilities of business organizations in Vietnam may be different to those in developed countries. The way to practice CSR may differ from country to country due to the influence of social contexts. For example, Vietnamese organizations tend to donate to religious festivals and provide 13th month wages' rewards for employees around New Year because of social expectations (Tran and Jeppesen 2016). Social influences may be a motive affecting the implementation of CSR in developing countries such as Vietnam. Literature shows that little research on CSR in SMEs has been conducted (e.g., Tilley 2000; Jenkins 2006; Morsing and Perrini 2009). The reasons for this relate to the importance of large organizations in the economy (Tilley 2000). Researchers may think that SMEs can use the theories and practices of CSR that were drawn from the study findings of large enterprises. However, the characteristics of SMEs are different to those of large ones (Spence and Rutherford 2003). For example, in SMEs, in most cases, ownership and management are not separate, so owners have the power to decide all business activities based on their personal preferences (Spence 1999; Perez-Sanchez et al. 2003; Davies and Crane 2010). The acceptance of putting CSR in SMEs can depend on the opinions and understandings of owners/managers about CSR, and this issue needs more research. Another SME issue relates to the lack of financial resources (Kusyk and Lozano 2007; Sweeney 2007), which means that SMEs may not focus on investment into CSR immediately—this study examines this.

The concept of CSR has been present in Vietnam since 2002, when it was introduced by international organizations such as the World Bank (Tran and Jeppesen 2016). Research about CSR in Vietnam also began at the same time. My literature review shows that there is little research on it, particularly relating to SMEs. Most research focuses on customers' perception of CSR (see, e.g., Bui 2010; Thi and Van 2016; Tran et al. 2017; Van et al. 2019). In other words, it explored how Vietnamese customers understand CSR. Findings showed that the concept of CSR is still new to research participants. In addition, many studies in Vietnam used the CSR dimensions by Carroll (1991) such as economic, legal, ethical, and philanthropic responsibilities (see, e.g., Bui 2010; Thi and Van 2016). They have not pointed out drivers such as the internal environment of SMEs that impact the implementation of CSR in firms—this is explored in our research.

Our review on the available literature shows that research about CSR in Vietnam usually studies the views of consumers, rather than managers, especially relating to SMEs (see, e.g., Bui 2010; Thi and Van 2016; Tran et al. 2017; Van et al. 2019). Collecting CSR opinions by customers may be easier than that by managers—this implies that our research, which studies manager perspectives, may be invaluable for literature. The study by Tran and Jeppesen (2016) was one of the few exploring the voice of managers about CSR in SMEs in Vietnam. Tran and Jeppesen (2016) interviewed 20 managers and 125 workers in 20 Vietnamese textile, garment, and footwear firms. Their study explored the way to practice CSR in firms, such as physical environment, working conditions, wages, and benefits. Their



research findings showed that CSR was not implemented in firms because financial resources were limited, and managers and workers did not understand the concept of CSR.

Nguyen and Pham (2016) are other researchers studying factors influencing the implementation of CSR in Vietnamese firms. Four factors, e.g., the regulatory system, knowledge of CSR, capital and human resources, were used for the test in their research. They sent a questionnaire to 207 firms in different business fields. Their research results showed that the only two factors affecting the way the firms practiced CSR are capital and human resources. As can be seen, SMEs in Vietnam may not be ready for the adoption of CSR due to cash limitations; however, in the industry of textiles and garments, they may be, due to the pressure of international partners or a competitive environment. Factors impacting the adoption of CSR in firms in Vietnam such as a competitive context, social influences, the understanding of managers about CSR and the internal environment of firms have not been the focus in these studies as of yet. Our research examines them from the view of managers who can play a decisive role in putting CSR in firms' developmental strategy.

As analyzed above, in order to study factors affecting the engagement of CSR in SMEs, the hypotheses were developed as follows:

**H1.** *An understanding of managers about CSR has a significant impact on the implementation of CSR in SMEs.*

**H2.** *The external and internal environment of SMEs positively affects their engagement in CSR. The environment consists of a competitive context, social influences, and the internal environment of companies.*

## 3. Method

Pilot tests and face-to-face interviews with four Vietnamese managers in the textile and garment industry were conducted to adjust, enhance, and validate the observed variables and the suggested measurement scales. Based on the feedback from the participants, all observed variables were corrected (see also Table 1).

Quantitative research was a survey on 330 top-, middle-, and low-level managers in different SMEs in Vietnam. We chose the firms based on the list of registered SMEs in the textile and garment industry provided by the Ho Chi Minh City Tax Department. The managers were not in the same companies. The surveys in Vietnamese, undertaken from March to October in 2017, were used for the study. After the returned questionnaires were reviewed and the invalid ones were eliminated, 250 valid answered questionnaires were coded to SPSS 23.0 for analysis purposes.

We measured CSR from the managers' perspectives using 5 factors with 25 observed items: the understanding of managers about CSR (4 variables: LĐ1, LĐ2, LĐ3, and LĐ4), the internal environment of companies (6 variables: MTNB1, MTNB2, MTNB3, MTNB4, MTNB5, and MTND6), competitive context (6 variables: MTCT1, MTCT2, MTCT3, MTCT4, MTCT5, and MTCT6), social influences (4 variables: MTVM1, MTVM2, MTVM3, and MTVM4), and the implementation of CSR in SMEs (5 variables: CSR1, CSR2, CSR3, CSR4, and CSR5). These variables were employed using research by Burke and Logsdon (1996), Spence et al. (2003), Murrillo and Lozano (2006), Porter and Kramer (2007), and Darnall et al. (2010). These were translated into Vietnamese. The scale was designed as a 5-point Likert scale, from 1—Totally disagree to 5—Totally agree (see Table 1). The study used Cronbach's alpha to test the reliability of the scale, exploratory factor analysis (EFA) to identify relationships between measured variables, and Pearson to study the correlation coefficient between dependent and independent variables (see also Section 4). The language used for the data collection and data analysis was Vietnamese. Thus, the codes for the items were named based on the Vietnamese language (Table 1). The findings of the research were translated into English for this paper.

**Table 1.** Descriptive statistics.

| Code | Observed variable (*n* = 250) | N | Min | Max | Mean | Std. Deviation |
|------|-------------------------------|---|-----|-----|------|----------------|
| | The implementation of CSR in SMEs | | | | | |
| CSR1 | CSR is embedded in firms' business responsibilities. | 250 | 2.0 | 5.0 | 3.35 | 0.691 |
| CSR2 | CSR has brought specific benefits for companies. | 250 | 2.0 | 5.0 | 3.46 | 0.706 |
| CSR3 | Firms voluntarily practice CSR. | 250 | 2.0 | 5.0 | 3.42 | 0.714 |
| CSR4 | Practicing CSR is a part of the strategy and business plan of companies. | 250 | 2.0 | 5.0 | 3.48 | 0.695 |
| CSR5 | Firms have an annual report of CSR for stakeholders as required. | 250 | 2.0 | 5.0 | 3.33 | 0.675 |
| | The understanding of managers about CSR | | | | | |
| LĐ1 | You understand and participate in CSR programs in your company. | 250 | 2.0 | 5.0 | 3.54 | 0.910 |
| LĐ2 | You participate in evaluating the benefits of CSR in your company. | 250 | 2.0 | 5.0 | 3.51 | 0.870 |
| LĐ3 | You understand that stakeholders (shareholders, employees, competitors, suppliers, customers) have an influence on the strategy and plans of your company. | 250 | 2.0 | 5.0 | 3.46 | 0.957 |
| LĐ4 | You think that companies need to have responsibilities to their stakeholders. | 250 | 2.0 | 5.0 | 3.48 | 0.875 |
| | The internal environment of SMEs | | | | | |
| MTNB1 | Employees put pressure on your companies in terms of implementing CSR in practice. | 250 | 2.0 | 5.0 | 3.57 | .839 |
| MTNB2 | Following labor laws and ensuring fair working conditions for employees have an influence on the practice of CSR in your company. | 250 | 2.0 | 5.0 | 3.53 | .870 |
| MTNB3 | Following health and safety policies and making a report about these have an influence on the practice of CSR in your company. | 250 | 2.0 | 5.0 | 3.72 | 1.041 |
| MTNB4 | Policies to motivate employees to improve practical skills and competencies have an influence on the practice of CSR in your company. | 250 | 2.0 | 5.0 | 3.61 | .805 |
| MTNB5 | Listening to employees' opinions about important issues has an influence on the practice of CSR in your company. | 250 | 2.0 | 5.0 | 3.58 | 0.857 |
| MTNB6 | CSR is a part of the marketing strategy of your company. | 250 | 2.0 | 5.0 | 3.56 | 0.891 |
| | Competitive context | | | | | |
| MTCT1 | Customers have an influence on the strategy, plans and decisions of your company. | 250 | 2.0 | 5.0 | 3.49 | 0.879 |
| MTCT2 | International partners have important requirements to the practice of CSR in your company. | 250 | 2.0 | 5.0 | 3.50 | 0.970 |
| MTCT3 | Suppliers and business partners have an influence on the strategy, plans and decisions of your company. | 250 | 2.0 | 5.0 | 3.52 | 0.906 |
| MTCT4 | Policies and standards of the industry and market about CSR have an influence on company. | 250 | 2.0 | 5.0 | 3.52 | 0.915 |
| MTCT5 | Suppliers have an influence on the plans and decisions of CSR implementation in your company. | 250 | 2.0 | 5.0 | 3.66 | 0.727 |
| MTCT6 | Formal and informal surveys of customer satisfaction have an influence on the practice of CSR in your company. | 250 | 2.0 | 5.0 | 3.51 | 0.919 |
| | Social influences | | | | | |
| MTVM1 | Local communities have an influence on the strategy, plans and decisions of the CSR implementation in your company. | 240 | 2.0 | 5.0 | 3.55 | 0.953 |
| MTVM2 | Governmental agencies have an influence on the strategy, plans and decisions of the CSR implementation in your company. | 250 | 2.0 | 5.0 | 3.55 | 0.914 |
| MTVM3 | Associations and non-governmental organizations have an influence on the strategy, plans and decisions of the CSR implementation in your company. | 250 | 2.0 | 5.0 | 3.59 | 0.906 |
| MTVM4 | Society has an influence on the strategy, plans and decisions of the CSR implementation in your company. | 250 | 1.0 | 5.0 | 3.56 | 0.944 |

## 4. Research Findings

### 4.1. Descriptive Statistics

After we deleted items with missing data, our final sample of 250 respondents was used for analysis. Respondents were top managers (46.4%), middle managers (34.4%), and low-level managers (19.2%). They were from companies with fewer than 50 employees (4.8%), between 50 and 200 employees (45.6%), between 200 and 300 employees (42.8%), and over 300 employees (6.8%). In terms of charter capital, the majority of companies (65.6%) were below \$430,000; the remaining ones (34.4%) were from \$430,000 to \$4,300,000. Almost all respondents were managers with more than three years' experience in the same position.

The statistical results described 25 observed items. The average value for the 25 observed items ranges from 3.33 to 3.72. Means are different between different components (see Table 1).

### 4.2. Cronbach's Alpha and EFA Analysis

The scale's reliability was tested via Cronbach's alpha. Cronbach's alphas of all variables were acceptable (>0.7). Item-total correlations were good (>0.3) (Table 2). Item reliability was also assessed by examining the factor loadings of each item with its respective latent variables. Kaiser–Meyer–Olkin (KMO) and Barlett's test was 0.852 for the all factors (sig. = 0.000 < 0.005). All observed items were eligible to be continued with EFA analysis.

**Table 2.** Cronbach's alpha of observed variables.

| Observed Variable | Scale Mean if Item Deleted | Scale Variance if Item Deleted | Item-Total Correlations | Cronbach's Alpha if Item Deleted |
|---|---|---|---|---|
| Cronbach's Alpha The understanding of Managers about CSR (LĐ): 0.733 | | | | |
| LĐ 1 | 10.45 | 4.280 | 0.571 | 0.646 |
| LĐ 2 | 10.48 | 4.684 | 0.482 | 0.697 |
| LĐ 3 | 10.53 | 4.154 | 0.561 | 0.652 |
| LĐ 4 | 10.50 | 4.661 | 0.485 | 0.695 |
| Cronbach's Alpha The Internal Environment of companies (MTNB): 0.812 | | | | |
| MTNB1 | 18.01 | 10.659 | 0.591 | 0.779 |
| MTNB2 | 18.05 | 10.335 | 0.627 | 0.771 |
| MTNB3 | 17.85 | 9.532 | 0.619 | 0.773 |
| MTNB4 | 17.96 | 11.368 | 0.417 | 0.802 |
| MTNB5 | 18.00 | 10.622 | 0.580 | 0.781 |
| MTNB6 | 18.01 | 10.598 | 0.553 | 0.787 |
| Cronbach's Alpha Competitive Context (MTCT): 0.818 | | | | |
| MTCT 1 | 17.72 | 10.948 | 0.547 | 0.796 |
| MTCT 2 | 17.71 | 10.134 | 0.619 | 0.781 |
| MTCT 3 | 17.69 | 10.792 | 0.552 | 0.795 |
| MTCT 4 | 17.69 | 10.527 | 0.592 | 0.786 |
| MTCT 5 | 17.55 | 11.549 | 0.571 | 0.794 |
| MTCT 6 | 17.71 | 10.400 | 0.616 | 0.781 |

**Table 2.** *Cont.*

| Observed Variable | Scale Mean if Item Deleted | Scale Variance if Item Deleted | Item-Total Correlations | Cronbach's Alpha if Item Deleted |
|---|---|---|---|---|
| Cronbach's Alpha Social Influences (MTVM): 0.744 | | | | |
| MTVM 1 | 10.70 | 4.871 | 0.483 | 0.716 |
| MTVM 2 | 10.69 | 4.680 | 0.580 | 0.660 |
| MTVM 3 | 10.66 | 4.909 | 0.517 | 0.696 |
| MTVM 4 | 10.69 | 4.609 | 0.570 | 0.666 |
| Cronbach's Alpha The Implementation of CSR in SMEs (CSR): 0.850 | | | | |
| CSR 1 | 9.72 | 5.301 | 0.710 | 0.806 |
| CSR 2 | 9.61 | 5.242 | 0.685 | 0.813 |
| CSR 3 | 9.66 | 5.472 | 0.636 | 0.826 |
| CSR 4 | 9.60 | 5.718 | 0.575 | 0.842 |
| CSR 5 | 9.76 | 5.519 | 0.705 | 0.809 |

EFA analysis was conducted to examine whether the items produced proposed factors, and if the results support the proposed five-factor solution. According to statistics results, all variables were grouped into five components at eigenvalue >1 and a cumulative sum of squared loadings >50%. All factors loadings were higher than 0.5 except MTCT5, therefore, this observed variable was removed from the analysis. The five groups were named internal environment, competitive context, managers, social influences, and CSR. Hence, the measurement scale meets the satisfactory level of reliability and validity.

*4.3. Correlation and Regression Analysis*

The Pearson's analysis was used to analyze the linear correlation between the dependent variable and independent variables. The correlation number had to be between –1 and 1 (IBM 2016). Table 3 shows that the variables are linearly related. The lowest correlation with the dependent variable is MTNB (0.402), whereas the highest is MTCT (0.681) (see Table 3). In terms of the regression equation, the adjusted R square (0.652) in Table 4 explains that the independent variables actually affect the dependent variable. The ANOVA test (Table 5) shows that the survey results are significant.

**Table 3.** Correlations.

| | | CSR | LĐ | MTNB | MTCT | MTVM |
|---|---|---|---|---|---|---|
| **CSR** | **Pearson Correlation Sig. (2-tailed)** | **1** | | | | |
| LĐ | Pearson Correlation | 505 ** | 1 | | | |
| | Sig. (2-tailed) | 0.000 | | | | |
| MTNB | Pearson Correlation | 0.402 ** | 0.329 ** | 1 | | |
| | Sig. (2-tailed) | 0.000 | 0.000 | | | |
| MTCT | Pearson Correlation | 0.681 ** | 0.386 ** | 0.257 ** | 1 | |
| | Sig. (2-tailed) | 0.000 | 0.000 | 0.000 | | |
| MTVM | Pearson Correlation | 0.611 ** | 0.404 ** | 0.383 ** | 0.343 ** | 1 |
| | Sig. (2-tailed) | 0.000 | 0.000 | 0.000 | 0.000 | |

** Correlation is significant at the 0.01 level (2-tailed).

**Table 4.** Model summary.

| Model | R | R Square | Adjusted R Square | Std. Error of the Estimate |
|-------|---|----------|-------------------|----------------------------|
| 1 | 0.811 [a] | 0.658 | 0.652 | 0.32987 |

[a] Dependent Variable: CSR.

**Table 5.** ANOVA.

| | Model | Sum of Squares | df | Mean Square | F | Sig. |
|---|-------|----------------|----|-------------|---|------|
| 1 | Regression | 51.244 | 4 | 12.811 | 117.733 | 0.000 [b] |
| | Residual | 26.660 | 245 | 0.109 | | |
| | Total | 77.904 | 249 | | | |

[b] Predictors: (Constant), MTVM, MTCT, MTNB, LĐ.

*4.4. Hypothesis Test*

Linear regression was conducted to test research hypotheses and the results are presented in Tables 4–6. The results show that the model is consistent with the data set (Sig. = 0.000 < 0.05) and CSR factors explain 65.2 percent of the implementation of CSR in SMEs (see Table 4). The VIFs (variance inflation factors) <2 indicate that there is no multi-collinearity issue in the data set. Table 6 shows that the four independent variances (LĐ, MTNB, MTCT, and MTVM) with sig. <0.05 are linearly related with the dependent variance (CSR). Research hypotheses indicate that H1 and H2 are accepted at the significance level of 1 percent. This means that the understanding of managers about CSR, and the external and internal environment of SMEs positively affect their engagement in CSR.

**Table 6.** Coefficients.

| Model | Unstandardized Coefficients | | Standardized Coefficients | t | Sig. | Collinearity Statistics | |
|-------|------|-----------|------|---|------|-----------|-----|
| | B | Std Error | Beta | | | Tolerance | VIF |
| Constant | 0.257 | 0.157 | | 1.642 | 0.102 | | |
| LĐ | 0.118 | 0.036 | 0.142 | 3.274 | 0.001 | 0.744 | 1.344 |
| MTNB | 0.105 | 0.036 | 0.119 | 2.871 | 0.004 | 0.810 | 1.234 |
| MTCT | 0.393 | 0.034 | 0.478 | 11.451 | 0.000 | 0.802 | 1.246 |
| MTVM | 0.276 | 0.035 | 0.344 | 7.923 | 0.000 | 0.739 | 1.353 |

**5. Discussion**

The research findings show that all proposed hypotheses were accepted. Impacts on the implementation of CSR in SMEs, are the understanding of managers about CSR, competitive context, social influences, and the internal environment of companies. Table 3 shows that MTCT and MTVM are the two strongest correlations with the dependent variance (CSR) compared to the others. Respondents highly value the relationships between SMEs' engagement in CSR, and competitive context and social influences (Table 6). The research hypothesis that the external environment (competitive context and social influences) positively affects SMEs' implementation of CSR was thus confirmed.

Table 3 also indicates that the correlations of LĐ and MTNT with the dependent variance were strong. The standardized coefficient of LĐ and MTNT (Table 6) showed significant relationships with the CSR variance. The proposed hypotheses about the influence of managers' CSR understanding and the internal environment of companies on the implementation of CSR in SMEs were confirmed.

The research findings showed that competitive context (observed variable MTCT) was the strongest driver to impact the strategy, plans, and decisions around CSR in SMEs (see Table 1, Table 3, and Table 6). This is a reason for SMEs to engage in CSR in order to reach their own business objectives.

The context here relates to customers, international partners, suppliers, business partners, policies and standards of the industry and market, and formal and informal surveys of customers. The large majority of goods (90%) in Vietnam's textile and garment industry are exported to other destinations such as the United States, Japan, and Europe (Song 2018; Duyen 2019). Thus, the expectations of external stakeholders, particularly international customers and business partners, may be important for companies in Vietnam in terms of the international standards of CSR and the industry. The pressure of shipment returns by European partners such as in 2018 (Bach 2019) can be an example to answer why competitive context is the most important impact on adopting CSR in companies. This finding is different to other research on CSR in SMEs (e.g., De Kok and Uhlaner 2001)—their research said that SMEs are often challenged from local rather than international markets and deal with less stakeholder stress. The difference can be due to the export market-specific characteristics of the industry in Vietnam. Another interesting point here is that SMEs have to put CSR in their business plans immediately because of stakeholder pressure and competitive context. Although SMEs have financial limitations compared to large enterprises (Kusyk and Lozano 2007; Sweeney 2007), they will adopt CSR soon owing to business requirements and stakeholders' satisfaction.

Social influences are ranked second among the four drivers (Table 1, Table 3, and Table 6). These are local communities, governmental agencies, associations, nongovernmental organizations, and society. This finding implies that there are requirements of society toward SMEs, meaning that companies are dependent on the society for their existence and developments. SMEs have to focus on social and political issues in their business activities. Vietnam is facing many pollution problems (World Health Organization 2018), so the society and local communities have put pressure on business companies due to fears around air and water pollution (see, e.g., Tan 2015; Le 2016; Ortmann 2017; Vinh 2019). Many environmental protest campaigns and movements have been organized by local people for many years. The campaigns led to the attention of governmental representatives in provinces and cities of Vietnam. Some provinces have thus established regulations on environmental issues. For example, Hue, a city in Central Vietnam, has a regulation on protecting the environment in the textile and garment industry (People's Committee of Hue 2018).

The understanding of managers about CSR is ranked third in the four factors (Table 6). Their understanding is about the influence of stakeholders on the strategy and plans of companies (Table 1). The respondents also agreed that participation in CSR programs and evaluating the benefits of CSR in companies is important for managers. The finding shows that there is a relationship between the implementation of CSR in companies and the understanding of managers about CSR (Table 6). This implies that the more managers understand CSR, the more they put CSR in their strategy and business plans (see also Table 1). In SMEs, decision-making tends to be based on the personal choices of managers (Spence 1999; Perez-Sanchez et al. 2003), so their understanding of CSR can support adopting CSR more directly and quickly than large companies do. This finding can help to consider that the CSR theories, which can successfully be applied to large enterprises, may need to be adjusted slightly for SMEs.

The last factor affecting the implementation of CSR in business is the internal environment of SMEs (Table 6). The managers in this study agreed that employees put pressure on their companies in terms of implementing CSR (Table 1). They also said that companies need to follow labor laws, health and safety policies, policies to motivate employees to improve practical skills and competencies, ensure fair working conditions for employees, and listen to employees' opinions. These issues have an influence on the strategy and plans of SMEs. As analyzed above, most companies in the textile and garment industry in Vietnam are exporters. Thus, they need to follow the standards of the CSR of countries where they supply goods. The majority of garment and textile goods are exported to other countries (Song 2018), therefore, international CSR standards (e.g., International Standards ISO 26000), which emphasize issues relating to employees, employment relationships, and labor practice, are all issues that the managers in this study were aware of. Moreover, Vietnam has recently had many protest campaigns relating to working conditions and employee wages (see, e.g., Dinh and Nguyen 2018).

For example, on 24 March 2018, thousands of workers in Pou Chen Corporation in the Dong Nai province, were on strike against the new salary policy that had been set by the company (Le 2018). Thus, internal environment is one of the drivers that force SMEs to engage in CSR.

Our research findings were different compared to those of other studies. For example, Tran and Jeppesen (2016) concluded that: "the CSR practices are not implemented because the business case for CSR in SMEs cannot be identified or because the western concept of CSR is not understood by managers and workers" (p. 605). However, our research found that SMEs adopt CSR due to the high expectations of stakeholders. The difference can be because of the research samples and time taken for data collection, even though the two studies explored CSR in the same field: the textile and garment industry. Tran and Jeppesen (2016) studied 20 managers in 20 firms, whereas our research surveyed 250 managers in 250 SMEs. Moreover, our research collected the data in 2017, while Tran and Jeppesen did theirs in 2011. This explains that the concept of CSR may now become popular in Vietnam and managers may recognize the importance of CSR for their business. In addition, our research found that competitive context, social influences, the understanding of managers about CSR, and the internal environment of SMEs are the four drivers influencing the implementation of CSR in SMEs. However, Nguyen and Pham (2016) said that SMEs only practice CSR if they have the cash and human resources. This implies that SMEs may not want to implement CSR due to the limitation of resources. Nevertheless, they can be ready to do it due to the pressure of international partners, competitive environment, local community, and society.

## 6. Conclusions

The relationship between the environment of SMEs and their engagement in CSR was tested in this research. The research also tested the link between the understanding of managers about CSR and the implementation of CSR in SMEs. The findings provide evidence of manager awareness about CSR practice in Vietnam. The research results indicated that competitive context, social influences, the internal environment of SMEs, and the understandings of managers about CSR are the four drivers of CSR. In the four drivers, competitive context has the strongest impact on adopting CSR in SMEs. This finding is different owing to the export market-specific characteristics of the textile and garment industry in Vietnam. The study results imply that stakeholder pressure influences companies due to the high expectations from international customers and partners, employees, local communities, and society. Another interesting finding relates to the characteristic of SMEs—personal decision making by managers/owners—in that the more managers/owners understand CSR and its benefits, the more CSR is incorporated in their business strategies. Although SMEs have financial limitations compared to large enterprises, SMEs can immediately add CSR to their business agenda because of stakeholder pressure and competitive context. This study provides evidence for the development of the theory of CSR that needs to consider more SMEs.

This study was conducted in Vietnam, where scholarship is emerging. This contributes to a broader understanding of CSR in SMEs, especially in the developing world. The research is also significant because it explores SMEs' engagement in CSR in a country where CSR is new to the public. Another contribution of this study relates to the research sample. The study collected data from the perspectives of SMEs, not those of large enterprises. In other words, our research findings reflect the view of SME managers who have an important role in making decisions about why firms need to add CSR to their business strategy. Finally, this research provides empirical evidence from the textile and garment industry. SMEs may have resource limitations, but they are ready to practice CSR due to the pressure of a competitive context and social influences.

There are limitations in this study that need to be considered in future research. Firstly, this study was conducted only in the textile and garment industry. Thus, generalizability of the research results is limited. Research in future should explore other industries so that it can discover similarities and differences among them. Secondly, our research surveyed SMEs only. Future studies need to examine large companies in order to compare the practice of CSR between SMEs and large enterprises in the

developing world, such as in Vietnam. Finally, future studies should be cross-country to develop a better understanding of the use of CSR theories in different countries.

**Author Contributions:** Conceptualization, L.T.-H.V.; methodology, A.P.N.; software, A.P.N.; formal analysis, L.T.-H.V. and A.P.N.; investigation, A.P.N.; resources, L.T.-H.V.; data curation, A.P.N. and L.T.-H.V.; writing—original draft preparation, L.T.-H.V.; writing—review and editing, L.T.-H.V.

**Funding:** This research received no external funding.

**Acknowledgments:** We are grateful to the three anonymous referees for their constructive comments. We also thank the participants at the 3rd Vietnam's Business and Economics Research Conference VBER2019 (Ho Chi Minh City Open University, Vietnam, 18-20 July 2019) for their helpful suggestions. The authors wish to acknowledge financial supports from Ho Chi Minh City Open University. The authors are solely responsible for any remaining errors or shortcomings.

**Conflicts of Interest:** The authors declare no conflicts of interest.

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
