# Peer review of "Corporate Social Responsibility and SMEs in Vietnam: A Study in the Textile and Garment Industry"

_jrfm, doi:10.3390/jrfm12040174_

Round 1

Reviewer 1 Report

- The literature review about CSR in SMEs needs further details. In fact, the attention paid to large firms and Vietnam's firms is higher than the attention paid to SMEs. This integration is relevant to extend the scientific soundness of your results. 

On this point, you can analyse the following papers:

Caputo, F., & Pizzi, S. (2019). Ethical firms and web reporting: Empirical evidence about the voluntary adoption of the Italian “legality rating”. Int. J. Bus. Manag, 14, 36-45.

Lythreatis, S., Mostafa, A. M. S., & Wang, X. (2019). Participative leadership and organizational identification in SMEs in the MENA Region: Testing the roles of CSR perceptions and pride in membership. Journal of Business Ethics, 156(3), 635-650.

Stekelorum, R., Laguir, I., & Elbaz, J. (2019). Cooperation with international NGOs and supplier assessment: Investigating the multiple mediating role of CSR activities in SMEs. Industrial Marketing Management

- I've got some questions about your methods:

1) what is your sampling strategy?How do you select your sample of SMEs?

2) You must provide a possible explanation about the correlation between CSR and MCTC. 

- The conclusions could be extended

Author Response

Dear the Reviewer

Thank you very much for the opportunity to revise and submit the paper to the Journal of Risk and Financial Management. Please be advised that all comments from the reviewers are now considered and incorporated in the revised version. More importantly, the paper will be professionally edited by the language editor recommended by the Reviewers to ensure consistency, clarity and coherence of information presented in the paper.

I would like to take this opportunity to thank you for your continuing support.

Best regards

Loan Van

Reviewer 2 Report

Review Report: Corporate Social Responsibility and SMEs in Vietnam: A Study in the Textile and Garment

Industry

The paper "Corporate Social Responsibility and SMEs in Vietnam: A Study in the Textile and Garment" would be of interest for the readers of the journal. I consider that the idea is interesting and worth doing research. The research topic is well explained as concerns its necessity. The paper is well organized, but it has several weaknesses that should be addressed.

Introduction and literature review should be improved. The introduction lacks scientific rigour. The introduction itself should provide clear description of the research area/problems, research gaps (short literature review indicating the problem in the literature showing – here shortly and more extensively in literature review section – what has been done and what you want to add in your paper) methodology of the paper (how you want to solve the problem) and your contribution to the literature. Make the introduction text clearer, focused and in line with the subject of your paper.  Moreover, the Authors should better address the context they want to investigate, i.e. why you focus on Vietnam and why Textile and Garment industry.

Possibly it would be good idea to add section (before Method section) presenting Textile and Garment industry in Vietnam in the context of CSR development. It would increase understanding of Vietnam context as well as Textile and Garment industry in Vietnam for foreign audience.

In fact, there is no literature review. Suggesting that there is little research is not enough. You should analyse referenced literature indicating what has been done to date, what variables they used, what sample they have chosen in their research and make a critical statement about what they have not done which you can add to the body of knowledge. This should be also presented in the introduction section but in a limited version (as mentioned above).

It is customary to add main results of the paper as well as an outline of the paper at the end of the introduction section.

I suggest authors to rewrite the introduction and conclusions sections following these instructions:

There is a discrepancy between the title and the content in section 2, since there is nothing about hypotheses. Hypotheses should be added to this section, clearly formulated and listed and be backed up with literature review and/or theory.

In Method section you should better backed up the choice of your variables. Was it your own choice or maybe you have based on previous literature? Moreover, have you regressed all 25 variables or summarize them to obtain 5 factors? In table 3 and 6 you have 5 variables (factors) not 25 items. This issue should be explained here. Additionally, please indicate regression method and present regression model.

Lines 167-179 should be moved to section 4.3. Correlation and Regression Analysis as they substantially relate to section 4.

In conclusion section there is nothing about research limitations and future research possibilities. Moreover, contribution of the paper should be the main part of conclusion section thus this item must be extended (at least contribution to literature, managers, policy makers).

Several minor comments to improve the paper:

- lines 95-97 “…ownership and management are not separated…” but there could be exceptions... so I suggest to change it on "in majority cases ownership and management are not separated... "

- in table 3 correlation LD-CSR possibly the point before correlation value is messing (.505)

Author Response

(The authors gave the same response as above.)

Reviewer 3 Report

The study is up-to-date and interesting.

Few suggestions which would improve the quality of the paper; changes which must be made before publication:

1. The structure of the paper needs attention and the usual rule
(introduction-rationale-need for the work/research questions,
background-literature review, approach-methods-research performed,
results, discussion and then conclusions/concluding remarks) should be
followed more closely to facilitate the flow of the paper. Please
develop further your discussion by drawing on relevant studies and perhaps in
relation with prior JRFM literature - develop further and expand your
final section of concluding remarks; incorporate research and policy
recommendations in the final conclusion section. Cite (primarily) in
these final-most critical sections of your manuscript relevant papers
published in the Journal you submitted your work to (in order to provide
some sort of continuity of the specific research string).

2. More references to recent and/or relevant literature/empirical studies
could increase the quality of the research paper and provide a much
clearer message to the reader - these may help you building your
discussion which needs to be extended. Please develop further your discussion by drawing on relevant studies and in relation with prior literature. Draw insights from the following papers & add those to your reference list:

Skouloudis, A...(2014). Exogenously-driven CSR: insights from the consultants' perspective. Business Ethics: A European Review, 23(3), 258-271.

... (2015). Priorities and perceptions
of corporate social responsibility: insights from the perspective of
Greek business professionals. Management Decision, 53(2), 375-401.

... (2015). Priorities and perceptions for corporate social responsibility: an NGO perspective. Corporate Social Responsibility and Environmental Management, 22(2), 95-112.   Halkos, G...(2017). Revisiting the relationship between corporate social responsibility and national culture: A quantitative assessment. Management decision, 55(3), 595-613.   Halkos, G...(2016). National CSR and institutional conditions: An exploratory study. Journal of Cleaner Production, 139, 1150-1156. Malesios, C...(2018). The impact of SME sustainability practices and performance on economic growth from a managerial perspective: Some modeling considerations and empirical analysis results. Business Strategy and the Environment, 960-972.  

Also consider the following keywords when revising your expanded discussion: Managerial perceptions/attitudes & national culture/national institutions, perceived role of ethics and social responsibility (PRESOR), SJ Vitell & JGP Paolillo, A Singhapakdi, enlightened self-interest

3. The introductory/opening section should communicate the research gaps aims, objectives and an short outline of the rest of the paper in order to
facilitate the flow of the study.

4. Final sections/Concluding remarks -  - elaborate on
research and policy recommendations in the conclusion section; authors
must elaborate more on what is their contribution to the literature as
well as on opportunities for future research. Questions that need to be
answered: Why your study is important? and how it extendso existing
knowledge on the issue/topic? Conclusions need to be written in a clear
and coherent manner and draw the main lessons from the paper. I suggest
you to concentrate on the description of the implications of the work,
the main findings and its replicability elsewhere. Furthermore,
limitations of the study need to be outlined to a greater extent, and so
are any potential connections between your study and specific aspects
of the Journal's scope.

5. Carefully check the references, so as to make sure they are all complete and follow the Guidelines to Authors.

6. Finally, when you submit the corrected version, please do check
thoroughly, in order to avoid grammar, syntax or structure/presentation
flaws. Make sure you retain a formal/academic-specific style of
presenting your work throughout the text - (if necessary) please seek
for professional English proofreading services or ask a native
English-speaking colleague of yours in order to refine and improve the
English in your paper

Author Response

(The authors gave the same response as above.)

Round 2

Reviewer 1 Report

Well done

Author Response

Thank you very much for your kindly help.

Reviewer 2 Report

I am happy with the changes made by the author. I accept the manuscript in present form.

Author Response

(The authors gave the same response as above.)

Reviewer 3 Report

Accept in present form

Author Response

(The authors gave the same response as above.)
